# Uniform Lithium Deposition Induced by ZnF_x_(OH)_y_ for High-Performance Sulfurized Polyacrylonitrile-Based Lithium-Sulfur Batteries

**DOI:** 10.3390/polym14214494

**Published:** 2022-10-24

**Authors:** Wanming Teng, Yanyan Li, Ting Ma, Xiuyun Ren, Ding Nan, Jun Liu, Xiaohu Wang, Qin Yang, Jiaojiao Deng

**Affiliations:** 1College of Chemistry and Chemical Engineering, Inner Mongolia University, Hohhot 010021, China; 2Inner Mongolia Key Laboratory of Graphite and Graphene for Energy Storage and Coating, School of Materials Science and Engineering, Inner Mongolia University of Technology, Hohhot 010051, China; 3State Key Laboratory of High-Performance Ceramics and Superfine Microstructures, Shanghai Institute of Ceramics, Chinese Academy of Sciences, 1295 Dingxi Road, Shanghai 200050, China; 4College of Materials Science and Engineering, Qingdao University, Qingdao 266071, China; 5Rising Graphite Applied Technology Research Institute, Chinese Graphite Industrial Park-Xinghe, Ulanqab 013650, China; 6Shenzhen Key Laboratory on Power Battery Safety and Shenzhen Geim Graphene Center, Tsinghua Shenzhen International Graduate School (SIGS), Shenzhen 518071, China

**Keywords:** ZnF_x_(OH)_y_ modification, current collector, Li-S battery

## Abstract

Lithium metal batteries are emerging as the next generation of high-density electrochemical energy storage systems because of the ultra-high specific capacity and ultra-low electrochemical potential of the Li metal anode. However, the uneven Li deposition on commercial Cu current collectors result in low Coulombic efficiencies (CEs) and poor cycle life. In this research, we proposed the modification of ZnF_x_(OH)_y_ on Cu foils to expand the lifespan. As-generated ZnLi alloy and LiF could promote uniform Li nucleation and deposition, thus resulting in an improved Li plating/stripping CE and extended cycle life. The Li-S battery with sulfurized polyacrylonitrile cathode and Li-ZnF_x_(OH)_y_@Cu anode (N/P ratio of 1.5:1) maintains 95% capacity after 60 cycles, proving the feasibility of ZnF_x_(OH)_y_@Cu for practical applications.

## 1. Introduction

Lithium-ion batteries (LIBs) have been dominating the electrochemical energy storage market due to their light weight, high energy density, and long service life [1,2]. However, the current commercial LIB cathode and anode materials are approaching their capacity limit and it is quite difficult to achieve further improvement. With the rapid development of electric vehicles and portable electronics, the large-scale development of LIBs is constrained, so the demand for high-density energy storage systems is increasingly urgent [3,4]. Lithium metal anodes (LMAs) have attracted extensive research attention because of their ultra-high specific capacity (3860 mAh g^−1^) and ultra-low electrochemical potential (−3.04 V vs. standard hydrogen electrode) [5,6]. Thus, as-fabricated lithium metal batteries (LMBs) are regarded as one of the most promising next-generation battery technologies.

The practical application of LMBs still has great challenges. LMAs are highly reactive with conventional liquid electrolytes and undergo continuous side reactions with liquid electrolytes to generate solid electrolyte interfaces (SEIs). During the lithium plating/stripping process, LMAs bear a large volume change, resulting in continuous formation and destruction of SEIs [7,8]. Moreover, the uneven deposition of Li is prone to the formation of Li dendrites, which can penetrate the separator and cause internal short circuits in the battery cell [9,10]. Therefore, low Coulombic efficiency (CE) and safety issues hinder the practical applications of LMBs. To address the above issues, researchers have proposed various strategies, such as the structural design of LMAs [11,12], current collector modifications [13,14], utilization of artificial SEIs [15,16] and electrolytes [17,18].

Copper foils, as anode current collectors, are important components of battery cells. However, the lithiophobic copper is not conducive to uniform nucleation and deposition of lithium metal [19], which will reduce the long-cycle stability of LMAs and shorten the cycle life of LMBs, especially in the case of limited lithium sources in battery cells (Figure 1a) [20,21]. Surface modification of Cu foils for lowering the energy barrier of Li metal nucleation and inducing uniform deposition of Li metal is an effective strategy to improve LMAs’ stability [22,23]. Jiang et al. constructed a three-dimensional (3D) Cu current collector coated with a dense nano-sized silver layer. The unique 3D structure not only alleviates the volume change, but also induces the epitaxial growth of Li deposition, and the large specific surface area of the 3D structure reduces the effective current density, thereby reducing the polarization of Li deposition or stripping [24]. Huang et al. used ultrathin Au films to form functional coating layers on copper foils. After lithiation, Au transformed into a Li_x_Au alloy with superior lithophilicity, which can reduce the nucleation barrier for Li deposition and induce dendrite-free Li metal deposition [25]. However, noble metals, such as Ag and Au, are not suitable for production on a large scale. Cost-inexpensive transition metal compounds such as ZnO [26,27], Cu_2_S [22], and NaMg(Mn)F_3_ [23] have also been utilized to modify Cu current collectors. Yet, the synthetic procedures are still complicated.

Herein, through an in situ hydrolysis reaction, uniform and dense ZnF_x_(OH)_y_ layers were coated on Cu current collectors to form ZnF_x_(OH)_y_@Cu. ZnF_x_(OH)_y_ can react with lithium to form Li-Zn alloy and LiF, which can reduce the nucleation energy barrier of lithium metal and accelerate the diffusion rate of Li^+^ on the surface of LMAs (Figure 1b) [28]. In addition, the ZnF_x_(OH)_y_-modified layer can also improve the interfacial stability and inhibit the side reaction between LMAs and the electrolyte, thereby improving the CE of lithium plating/stripping and prolonging the cycle life [29]. Lithium metal was electrodeposited on ZnF_x_(OH)_y_@Cu current collector, and the formed Li/ZnF_x_(OH)_y_@Cu anode was assembled with pPAN/SeS_2_ cathode to form a lithium–sulfur (Li-S) battery. Under the condition of high cathode mass loading (~7 mg cm^−2^), lean electrolyte (20 μL) and limited lithium source (N/P = 1.5), the as-fabricated Li-S cell shows excellent cycling stability. It maintains as high as 95% of the original specific capacity after 60 cycles, thus proving the great potential of ZnF_x_(OH)_y_@Cu for advancing practical applications of Li-S batteries.

## 2. Experimental

### 2.1. Materials Synthesis

#### 2.1.1. Materials

Zinc fluoride tetrahydrate (ZnF_2_·4H_2_O, 98%, Aladdin), anhydrous lithium nitrate (LiNO_3_, 99.9%, Alfa Aesar), lithium bis(trifluoromethanesulfonyl)imide (LiTFSI), 1,3 dioxolane (DOL) and 1,2-dimethoxyethane (DME) were all purchased from TCI. Selenium disulfide (SeS_2_) and polyacrylonitrile (PAN) were both purchased from Sigma-Aldrich. Lithium foil (Li), aluminum foil (Al), copper foil (Cu), separator (Celgard 2400), carboxymethyl cellulose (CMC), styrene–butadiene rubber (SBR), and carbon black SUPER C45 (SP) were purchased from Hefei Kejing Material Technology Co., Ltd.

#### 2.1.2. Preparation of ZnF_x_(OH)_y_@Cu

First, a plastic beaker was filled with 200 mL deionized water, and 3.04 g of ZnF_2_·4H_2_O particles were taken and dispersed in the above deionized water by ultrasonic stirring until the solution became clear. Then, 5 * 5cm copper foil was washed with 50 mL of 1 mol/L dilute hydrochloric acid to remove surface impurities, and added to the above ZnF_2_·4H_2_O aqueous solution, which was mechanically stirred at room temperature for 3 h. Finally, the modified copper foil was washed with deionized water and ethanol 3-5 times to remove the alkaline salt on the modified copper foil, and the product obtained was named as ZnF_x_(OH)_y_@Cu. After the experiment, some lime water was prepared. The hydrolyzed HF was neutralized with lime water to produce non-toxic calcium fluoride precipitation.

#### 2.1.3. Preparation of pPAN/SeS_2_

PAN and SeS_2_ powder (1:4 *w*/*w*) were sealed in glass containers and incubated at 380 °C under argon for 8 h. The samples were then placed in a porcelain boat, kept at 350 °C for 6 h under flowing nitrogen to remove excess sulfur and selenium, and then naturally cooled to obtain the pPAN/SeS_2_ product.

### 2.2. Characterization

Scanning electron microscope (SEM, HITACHI-SU8220) was used to observe the microstructure and element distribution of ZnF_x_(OH)_y_@Cu. Elemental analysis was performed on an energy dispersive X-ray spectroscopy (EDX) spectrometer connected to a HITACHI-SU8220. The phase composition of ZnF_x_(OH)_y_@Cu was further determined by X-ray diffraction (XRD, SmartLab 9 kW). The test range was 10~80°, the scanning rate was 5°/min, and the Cu target was used in a continuous scanning mode. To observe the morphological changes of Li deposition after cycling of Cu and ZnF_x_(OH)_y_@Cu. The cells were disassembled and the lithium-coated anodes were rinsed with DME to remove electrolyte residues and LiTFSI salts on Li/Cu and Li/ZnF_x_(OH)_y_@Cu, and then dried for 24 h before characterization for testing. All procedures were performed in an argon-filled glove box.

### 2.3. Electrochemical Measurements

All electrochemical properties were measured using CR2032 coin cells, which were assembled in an argon-filled glove box with both O_2_ and H_2_O below 0.1 ppm. To evaluate Li plating/stripping efficiency, Li/Cu and Li/ZnF_x_(OH)_y_@Cu half cells were assembled using Cu or ZnF_x_(OH)_y_@Cu as the working electrode (φ16 mm) and Li foil (φ15.5 mm) as the counter/reference electrode. The cells were first cycled five times at 50 μA in the voltage range of 0–1 V (vs. Li^+^/Li), followed by a long-term cycling test at a current density of 0.5 mA cm^−2^ and a lithium deposition capacity of 1 mAh cm^−2^. For the full cell test, the cathode was composed of pPAN/SeS_2_, SP, and CMC/SBR mixed in a mass ratio of 8:1:1, and the mass loading of the cathode was ~7 mg cm^−2^. The Li-deposited ZnF_x_(OH)_y_@Cu was used as the anode, which was assembled with the high-load cathode to form a full cell. The applied current density during full cell cycling was 0.1 A g^−1^, the working potential window was 1 to 3 V, and the utilized electrolyte volume was ~20 µL. The electrolyte used in the cells was the solution of 1 M LiTFSI dissolved in DOL and DME (*v*/*v* = 1:1) with 3.7 wt% LiNO_3_ as additive, and the separator was Celgard 2400.

## 3. Results and Discussion

The ZnF_x_(OH)_y_@Cu current collector was fabricated through in situ hydrolysis of ZnF_2_ copper foil: ZnF_2_ + H_2_O → ZnF_x_(OH)_y_ + HF (Figure 2a). The morphologies of the as-prepared ZnF_x_(OH)_y_@Cu current collectors were investigated by SEM. As shown in Figure 2b–d, after copper foil was soaked in ZnF_2_ aqueous solution for 3h, polyhedral ZnF_x_(OH)_y_ particles were uniformly coated over the entire copper surface. The bare copper foil and ZnF_x_(OH)_y_@Cu showed completely different colors; the surface of bare copper foil was bright orange-pink, while the surface of ZnF_x_(OH)_y_@Cu was covered with a dark red substance (Appendix A). The morphologies of the as-prepared ZnF_x_(OH)_y_@Cu current collectors were investigated by SEM, and the results are presented in Figure 2. As shown in Figure 2b, after copper foil was soaked in ZnF_2_·4H_2_O aqueous solution for 3h, ZnF_x_(OH)_y_ particles were uniformly coated on the whole copper surface. The locally enlarged topography in Figure 2c shows that the ZnF_x_(OH)_y_ particles are not uniform in size, but are regular polyhedrons in shape. Similarly, in Figure 2d, it can be observed that the polyhedral ZnF_x_(OH)_y_ particles effectively coat the copper foil and enhance the adhesion performance to the copper foil current collector without the formation of aggregates. Furthermore, the EDX mapping images clearly show that Zn and F elements are uniformly distributed on the ZnF_x_(OH)_y_@Cu current collector, demonstrating the homogeneity of the ZnF_x_(OH)_y_ coating (Appendix A). The phase composition of ZnF_x_(OH)_y_@Cu was further analyzed by XRD. There are no characteristic diffraction peaks of ZnF_x_(OH)_y_ in the spectrum, indicating that the deposited ZnF_x_(OH)_y_ films are amorphous (Appendix A).

To explore the electrochemical properties of ZnF_x_(OH)_y_@Cu, Li/ZnF_x_(OH)_y_@Cu half cells were assembled. Figure 3a demonstrates Coulombic efficiencies of Li plating/stripping on Cu and ZnF_x_(OH)_y_@Cu current collectors. At a current density of 0.5 mA cm^−2^ and areal Li deposition capacity of 1 mAh cm^−2^, the first Coulombic efficiencies of the anodes based on bare Cu and ZnF_x_(OH)_y_@Cu are 97.1% and 98.5%, respectively. The lower initial Coulombic efficiency is related to the formation of a solid electrolyte interphase (SEI) layer during the first cycle. For the SEI layer on the anode of the lithium metal battery, it is generally believed that the extremely reactive metal Li reacts with the anion in the electrolyte, and the reaction products (mostly insoluble) are deposited on the surface of lithium metal, forming a passivation film thick enough to prevent electrons from passing through. The formation of the SEI layer will consume active Li in the battery, resulting in low Coulombic efficiency. During cycling, the Coulombic efficiency of bare copper began to fluctuate slightly after 60 cycles, and the average Coulombic efficiency was 98.4% over 100 cycles. The fluctuating Coulombic efficiency of Li/Cu half cells proves that the surface SEI film was continuously collapsing and regenerating, consuming a large amount of lithium, while the average Coulombic efficiency of ZnF_x_(OH)_y_@Cu can be stabilized at about 98.8% over 100 cycles, implying that nucleation sites (ZnF_x_(OH)_y_) can promote uniform nucleation/deposition of Li and form a stable SEI layer.

The voltage profiles of Li plating/stripping during the first cycle of Cu and ZnF_x_(OH)_y_@Cu current collectors are shown in Figure 3b. Due to the lithiophobicity of Cu, the voltage drops to −124 mV (vs. Li^+^/Li) at the beginning of Li plating and then reaches a flat voltage plateau of −43 mV (vs. Li^+^/Li). The voltage difference between the bottom of the voltage dip and the plateau represents the overpotential barrier for heterogeneous Li nucleation deposition on the substrate [30]. The bare Cu foil exhibits a large nucleation overpotential of 54 mV. In contrast, the nucleation overpotential of ZnF_x_(OH)_y_@Cu is as low as 34 mV. These results indicate that the super-lithophilicity of ZnF_x_(OH)_y_@Cu helps to reduce the interface energy between Li and the deposition substrate and lower the heterogeneous nucleation barrier. Moreover, the ZnF_x_(OH)_y_ particles react with Li to form an favorable SEI layer, which induces uniform growth of Li metal, thereby suppressing the formation of Li dendrites. The morphological evolution of Li plating/stripping on the bare copper and ZnF_x_(OH)_y_@Cu were investigated by SEM. Appendix A shows the morphologies of bare copper and ZnF_x_(OH)_y_@Cu at a current density of 0.5 mA cm^−2^ and a plating capacity of 1 mAh cm^−2^ after 1 cycle of plating, showing the deposition of limited amounts of lithium. It can be observed that a very rough surface is displayed on the bare copper, and the Li deposition on bare copper exhibited highly porous and rough non-uniform surface morphology, on which Li particles, accompanied by a multitude of dead Li, were non-uniform in size. This moss-like Li morphology may lead to persistent side reactions with the electrolyte to accelerate Li and electrolyte consumption, whereas compared with uncontrolled growth of Li dendrites on the bare copper, the Li deposition shows uniform and dense morphology on the surface of ZnF_x_(OH)_y_@Cu, which is favorable for a good cycling stability. The improved Li deposition can be attributed to the fact that the lithiophilic ZnF_x_(OH)_y_ coating can provide multiple and uniform nucleation sites, which can effectively induce smoother Li deposition and inhibit the growth of Li dendrites, thereby improving the Coulombic efficiency and cycle life of the battery. Both of surface morphology and the morphological evolution of Li plating/stripping demonstrate the better performance of the surface-modified ZnF_x_(OH)_y_@Cu, which is consistent with the electrochemical data. Furthermore, the voltage profiles of the 2, 5, 10, 50, and 100 cycles of ZnF_x_(OH)_y_@Cu show that the lithium cycling presents stable polarization at a current density of 0.5 mA cm^−2^. The discharge–charge curves of Li/ZnF_x_(OH)_y_@Cu batteries almost overlap, again confirming the excellent cycling stability (Appendix A). The above results indicate that the modified Li/ZnF_x_(OH)_y_@Cu cell has a more stable electrochemical cycling behavior. To confirm the low polarization and cycling stability of ZnF_x_(OH)_y_@Cu, we examined the resistance changes of Li/Cu and Li/ZnF_x_(OH)_y_@Cu after 5 cycles using electrochemical impedance spectroscopy (EIS), to analyze internal resistance and interface stability. The radius of the depression in the high frequency region usually reflects the value of the charge transfer resistance (R_ct_). Compared with the bare copper current collector, the interfacial charge resistance value of the lithiated ZnF_x_(OH)_y_@Cu electrode is significantly lower, indicating that lithium ions form a stable and thin SEI layer on the surface of the ZnF_x_(OH)_y_@Cu electrode, and the lithium ions form a stable and thin SEI layer on the surface of the ZnF_x_(OH)_y_@Cu electrode. The kinetics of ions during cycling are improved and the transport in the SEI layer is faster. Therefore, the improved cycling stability and CE can be ascribed to more reaction sites provided by the lithiophilic ZnF_x_(OH)_y_ to induce uniform Li deposition, and the ZnF_x_(OH)_y_@Cu electrode has higher lithium ion diffusivity than the bare copper electrode (Appendix A).

N/P ratio refers to the areal capacity of the Li anode by that of the sulfur cathode [31]. A Li-S cell with maximized energy density should operate at an N/P ratio of 1 [32]. However, excessive Li is normally required to offset the Li loss from electrolyte consumption and SEI formation [33]. Here, we report polyacrylonitrile (PAN)/selenium disulfide (pPAN/SeS_2_) as a cathode material for Li-S batteries. The pPAN/SeS_2_ was synthesized from a mixture of SeS_2_ and PAN (4:1 by weight) under heat treatment at 380 °C. As shown in Figure 4a, SEM image shows the round morphology of pPAN/SeS_2_ particles, and most of the particles agglomerate into large clusters. The phase structure of pPAN/SeS_2_ was investigated by XRD. As shown in Figure 4b, the XRD pattern of PAN has a characteristic diffraction peak at 17°, which corresponds to the (110) crystal plane of the PAN crystal structure. The XRD pattern of SeS_2_ does not show a characteristic peak. In the XRD pattern of pPAN/SeS_2_, only a distinct broad peak appeared at 25°, demonstrating the amorphous phase of carbon. No diffraction peaks corresponding to SeS_2_ compounds were observed in the XRD pattern, indicating the formation of an amorphous structure in pPAN/SeS_2_ [34]. In order to construct a Li-S full battery with a thin lithium anode, based on the capacity of the highly loaded pPAN/SeS_2_ cathode, we calculated that when N/P = 1.5:1, the areal capacity of Cu or ZnF_x_(OH)_y_@Cu current collectors for lithium deposition is 6.0 mAh cm^−2^. Li/Cu and Li/ZnF_x_(OH)_y_@Cu cells were disassembled after deposition, and images of Cu or ZnF_x_(OH)_y_@Cu current collectors after Li deposition were compared (Appendix A); the ZnF_x_(OH)_y_@Cu current collector was fully covered by the deposited lithium while the bare Cu current collector was not, indicating the much higher Li deposition uniformity on ZnF_x_(OH)_y_@Cu.

In this research, Li-plated Cu and ZnF_x_(OH)_y_@Cu (the lithium deposition amount was 6.0 mAh cm^−2^) were fabricated as the anodes, and pPAN/SeS_2_ were used as the cathode. Se has better ionic/electronic conductivity than S, and can accelerate the conversion reaction kinetics, which has been demonstrated in previous research [35]. Li-S cells were constructed and tested under practical testing conditions of high cathode mass loading (~7 mg cm^−2^), lean electrolyte (E/S ratio of less than 3 μL mg^−1^), and low N/P ratio of 1.5:1. The cycling performance of both full cells were evaluated at a current density of 0.1 A g^−1^. As shown in Figure 4c, the capacity of the Li/Cu|pPAN/SeS_2_ full cell decreased as the cycling progressed, resulting in an 88% capacity retention after 60 cycles. The capacity decline of the Li/Cu|pPAN/SeS_2_ full cell may be due to the uneven distribution of lithium on the bare copper foil, and as the cycling continues, lithium dendrites accumulate on the surface of the bare copper foil, resulting in the decay of the active capacity. In contrast, the Li/ZnF_x_(OH)_y_@Cu|pPAN/SeS_2_ full cell maintained a discharge capacity of 519 mAh g^−1^ after 60 cycles with a capacity retention rate of 95%, which was significantly better than that of the Li/Cu|pPAN/SeS_2_ full cell, with a CE of ~100%. It can be seen that a uniform and stable Li-deposited interfacial layer will contribute to the cycling stability of the battery.

The discharge–charge curves of the Li/ZnF_x_(OH)_y_@Cu|pPAN/SeS_2_ full cell at 0.1 A g^−1^ are shown in Figure 4d. The Li/ZnF_x_(OH)_y_@Cu|pPAN/SeS_2_ full cell exhibits a very stable voltage profile during successive discharge–charge cycles. At the 50th cycle, the Li/ZnF_x_(OH)_y_@Cu|pPAN/SeS_2_ full cell showed a more stable voltage plateau and lower voltage polarization than the Li/Cu|pPAN/SeS_2_ full cell, indicating a significant kinetics improving (Appendix A). The above results indicate that the uniform Li layer in the full cell reduces the subsequent consumption of limited Li and electrolyte due to the stable interface and low Li nucleation barrier of ZnF_x_(OH)_y_@Cu. Therefore, the modified Li/ZnF_x_(OH)_y_@Cu|pPAN/SeS_2_ full cell can maintain high capacity retention, which is beneficial to prolong the cycle life of Li-S batteries.

## 4. Conclusions

In summary, uniform and dense ZnF_x_(OH)_y_ layers have been coated on Cu current collectors through in situ hydrolysis. ZnF_x_(OH)_y_ reacts with lithium to form Li-Zn alloy and LiF, which reduces the Li nucleation energy barrier and facilitates rapid Li^+^ diffusion at the surface of LMAs. As a result, the average Coulombic efficiency of ZnF_x_(OH)_y_@Cu can be improved to 98.8% over 100 cycles. Under practical testing conditions of high cathode mass loading (~7 mg cm^−2^), lean electrolyte (E/S ratio of less than 3 μL mg^−1^), and low N/P ratio of 1.5:1, the Li/ZnF_x_(OH)_y_@Cu|pPAN/SeS_2_ full cell shows a superior cycling stability, with a capacity retention ratio as high as 95% after 60 cycles. This work sheds light on the great potential of ZnF_x_(OH)_y_ modification on Cu foil for advancing practical applications of Li-S batteries.

## Figures and Tables

**Figure 1 polymers-14-04494-f001:**
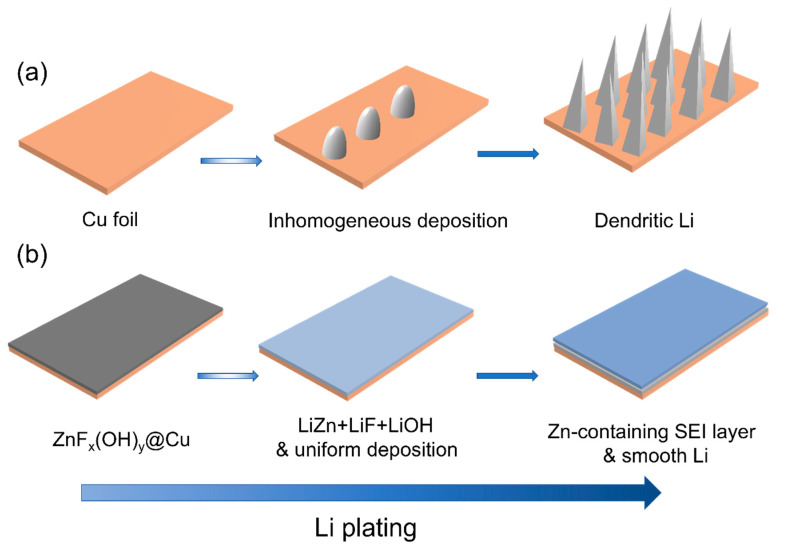
Schematics for the Li plating morphologies on different current collectors: (**a**) Behavior of Li nucleation and dendrite growth during Li plating on a bare Cu foil. (**b**) Upon Li plating on ZnF_x_(OH)_y_@Cu, the in situ-formed ZnF_x_(OH)_y_ layer can effectively guide the uniform deposition of Li via its lithophilic nature to form a stable solid-electrolyte interphase during cycling, leading to uniform distribution of Li^+^ ion flux and stable Li electroplating.

**Figure 2 polymers-14-04494-f002:**
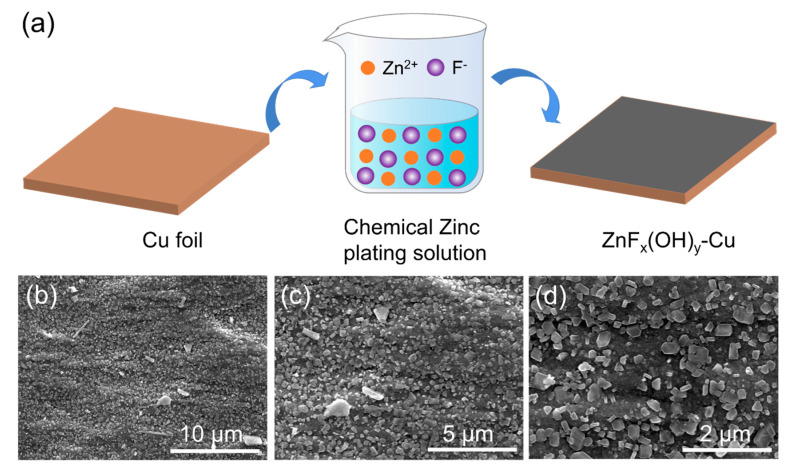
(**a**) Schematic diagram of the preparation of ZnF_x_(OH)_y_@Cu. Copper foil was surface-modified in ZnF_2_·4H_2_O aqueous solution. The copper foil undergoes in situ hydrolysis reaction in a chemical zinc plating solution to coat ZnF_x_(OH)_y_ particles on a conventional planar Cu current collector (referred to as ZnF_x_(OH)_y_@Cu). (**b**–**d**) SEM images of the ZnF_x_(OH)_y_@Cu-modified current collectors.

**Figure 3 polymers-14-04494-f003:**
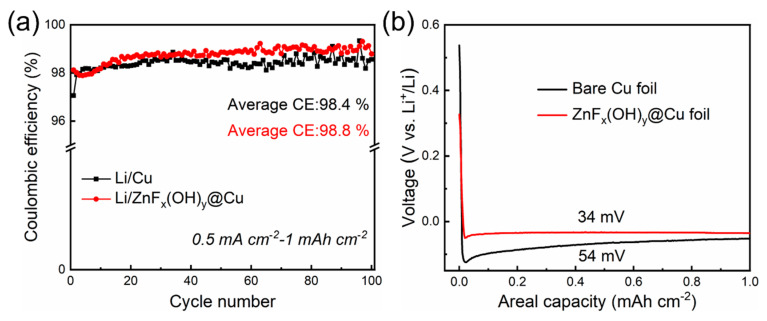
Cyclic stability of different anodes in the half cells with the ether electrolyte, at a current density of 0.5 mA cm^−2^ and a plating capacity of 1 mAh cm^−2^. (**a**) Coulombic efficiency of the anodes based on bare Cu and ZnF_x_(OH)_y_@Cu. (**b**) Electrochemical Li plating curves on bare Cu foil and ZnF_x_(OH)_y_@Cu at 0.5 mA cm^−2^ with a specific capacity of 1 mAh cm^−2^.

**Figure 4 polymers-14-04494-f004:**
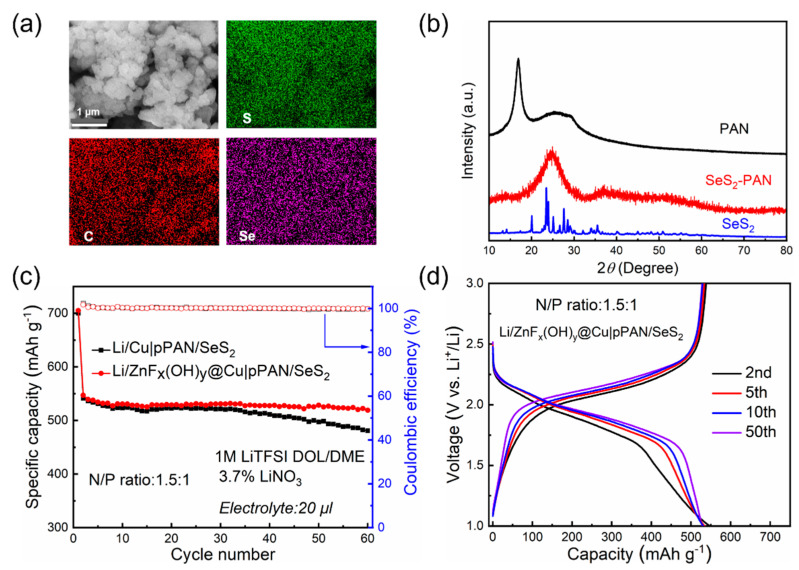
(**a**) SEM image and corresponding EDS element mapping (sulfur, carbon, selenium) of pPAN/SeS_2_ sample. (**b**) XRD patterns of PAN (black), pPAN/SeS_2_ (red), and SeS_2_ (blue) sample. (**c**) Cycling performance of Li/Cu|pPAN/SeS_2_ and Li/ZnF_x_(OH)_y_@Cu|pPAN/SeS_2_ full cell at a current density of 0.1 A g^−1^. (**d**) The discharge–charge curves of Li/ZnFx(OH)y@Cu|pPAN/SeS_2_ full cell at different cycles.

## Data Availability

Not applicable.

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
