# Peer review of "Uniform Lithium Deposition Induced by ZnFx(OH)y for High-Performance Sulfurized Polyacrylonitrile-Based Lithium-Sulfur Batteries"

_polymers, 2022, doi:10.3390/polym14214494_

Round 1

Reviewer 1 Report

The paper brings out a novel concept about modifying current collectors to improve the battery performance. However some points need further clarifications:

1. Zinc Fluoride poses potential health and environmental hazards. The authors are unclear about how the material is handled and how the experiment (step 2.1.2) is performed.

2. In figure 3(a), in spite of fluctuations of CE for the Li/Cu sample from around cycle 60, the CE's are almost similar. So does that mean the benefit lies pretty much in the stabilization of the SEI ?

3. In figure 4(c), the specific capacities are almost identical up to the 30th cycle and then fall away. It'd be interesting to see any capacity decline for the ZnFx(OH)y treated cell after about 100-150 cycles also.

Author Response

Reviewer 1#

The paper brings out a novel concept about modifying current collectors to improve the battery performance. However some points need further clarifications:

  1. Zinc Fluoride poses potential health and environmental hazards. The authors are unclear about how the material is handled and how the experiment (step 2.1.2) is performed.

Reply: Thanks for the reviewer’s comment. We agree with the reviewer that zinc fluoride poses potential health and environmental hazards. ZnF2 • 4H2O is stored in a dry container. For the lab-scale experiment, the amount of HF gas released is very tiny (bubbles can not be observed by naked eyes), the experiment is carried out in the fume hood, and the waste liquid was treated with lime water to remove the HF residual. Meanwhile, the operator wears protective mask, goggles, gloves and lab coat. If the production scale is magnified, a sealed reaction system and exhaust gas recovery system should be utilized.

As for the experiment step 2.1.2, a more specific operation process is described as follows and added in the revised manuscript: First, a plastic beaker was filled with 200 ml deionized water, and 3.04 g of ZnF2•4H2O particles were taken and dispersed in the above deionized water by ultrasonic stirring until the solution became clear. Then, 5*5cm copper foil was washed with 50 ml of 1 mol/L dilute hydrochloric acid to remove surface impurities, and added to the above ZnF2•4H2O aqueous solution, which was mechanically stirred at room temperature for 3 hours. Finally, the modified copper foil was washed with deionized water and ethanol for 3-5 times to remove the alkaline salt on the modified copper foil, and the product obtained was named as ZnFx(OH)y@Cu. After the experiment, some lime water was prepared. The hydrolyzed HF was neutralized with lime water to produce non-toxic calcium fluoride precipitation.

  1. In figure 3(a), in spite of fluctuations of CE for the Li/Cu sample from around cycle 60, the CE's are almost similar. So does that mean the benefit lies pretty much in the stabilization of the SEI ?

Reply: CE mainly depends on the composition of electrolytes including salts and solvents. In this research, the working electrode and reference electrode were tested in the same electrolyte. Therefore, CE values are quite similar in two systems. And we agree with reviewer that the stability of CE over cycling (with or without fluctuation) mainly depends on the stability of SEI. For the working electrode, ZnF(OH) nanocrystals promote uniform Li nucleation and deposition. Therefore, the stability of as-generated SEI is better than that in the reference electrode.

  1. In figure 4(c), the specific capacities are almost identical up to the 30th cycle and then fall away. It'd be interesting to see any capacity decline for the ZnFx(OH)y treated cell after about 100-150 cycles also.

Reply: Thanks for the reviewer’s comment. As the reviewer said, the capacity of both samples declined after 30 cycles, but sample ZnFx(OH)y@Cu declined slowly, while the battery capacity of pure copper foil declined rapidly. The reason for the decrease of the capacity of the two samples is that the lithium source is gradually consumed during cycling. However, the capacity of sample ZnFx(OH)y@Cu decreases slowly, because Zn can promote the uniform deposition of lithium metal, inhibit the formation of lithium dendrites and dead lithium, and thus reduce the loss of lithium during repeated cycles. In the anode-free cell, as Li source is gradually consumed, the specific capacity would decrease faster and faster, and finally the cell would fail.

Reviewer 2 Report

The paper is devoted to a new coating process for the anodic current collector and to a new formulation of the cathode material for Li-S batteries. The results are presented in a clear manner and no major changes are required in my opinion.

However I suggest to improve the introduction on Li-ion batteries citing more recent reviews (e.g. doi.org/10.1016/j.adapen.2021.100070).

Moreover, please improve the size and the quality of the EDS maps reported in figure 4a.

Author Response

Reviewer 2#

The paper is devoted to a new coating process for the anodic current collector and to a new formulation of the cathode material for Li-S batteries. The results are presented in a clear manner and no major changes are required in my opinion.

However I suggest to improve the introduction on Li-ion batteries citing more recent reviews (e.g. doi.org/10.1016/j.adapen.2021.100070).

Reply: We thank the reviewer for the valuable comment. The introduction has been improved in the revised manuscript. The recommended references have been cited.

Moreover, please improve the size and the quality of the EDS maps reported in figure 4a.

Reply: We thank the reviewer for the valuable comment. The quality of Figure 4a has been improved in the revised manuscript.

Reviewer 3 Report

"Uniform Lithium Deposition Induced by ZnFx (OH) y for High Performance Sulfurized Polyacrylonitrile-Based Lithium-Sulfur Batteries"

This research article is focused on the utilization of the metal anodes for the better performance of the Li-S batteries. In this research, the layer of ZnFx (OH) y over the current collector Copper (Cu) managed the uniform growth of Lithium instead of Li dendrite growth. Simple Hydrolysis and electrodeposition technique is used for preparation of anode materials for the Lithium-Sulphur (Li-S) batteries.

Major issues:

1. The authors explain the cost effectiveness along with scalability, as several chemicals involved, and gas flow is required for long hours?

2. The toxic and environmental impact of the precursors is not discussed.

3. The SEI layer formation is mainly described by the 1st charge and discharge cycle is not disclosed in both main article and supplementary Materials?

4. The difference in coulombic efficiency for bare Cu(98.4%) and modified Cu(98.8%) is not much different which is basic claim of the study?(line 195)

5. The authors explain the high performance based on 100 or 60 cycles and 519mAh/g on the other hand we have graphite(anode material) with thousand cycles with low cost?

Minor issues:

1. Cyclic voltammetry is not mentioned in the research which is important

and corelated with the charge and discharge profile?

2. The role of Se is not mentioned in the research?

Author Response

Reviewer 3#

"Uniform Lithium Deposition Induced by ZnFx (OH) y for High Performance Sulfurized Polyacrylonitrile-Based Lithium-Sulfur Batteries"

This research article is focused on the utilization of the metal anodes for the better performance of the Li-S batteries. In this research, the layer of ZnFx(OH)y over the current collector Copper (Cu) managed the uniform growth of Lithium instead of Li dendrite growth. Simple Hydrolysis and electrodeposition technique is used for preparation of anode materials for the Lithium-Sulphur (Li-S) batteries.

Reply: We thank the reviewers for their valuable comments.

Major issues:

  1. The authors explain the cost effectiveness along with scalability, as several chemicals involved, and gas flow is required for long hours?

Reply: The cost effectiveness along with scalability is for the ZnFx(OH)y@Cu anode current collector but not for the pPAN/SeS2 cathode. The cost of raw materials (ZnF2•4H2O) are low. The ZnFx(OH)y@Cu was fabricated by a facile room-temperature solution reaction, no gas flow is required during preparation. The process is simple and easy to control, which is suitable for mass production.

  1. The toxic and environmental impact of the precursors is not discussed.

Reply: Thanks for the reviewer’s valuable comment. We agree with the reviewer that zinc fluoride poses potential health and environmental hazards, and should be of great consideration. ZnF2 • 4H2O is stored in a dry container. For the lab-scale experiment, the amount of HF gas released is very tiny (bubbles can not be observed by naked eyes), the experiment is carried out in the fume hood, and the waste liquid was treated with lime water to remove the HF residual. Meanwhile, the operator wears protective mask, goggles, gloves and lab coat. If the production scale is magnified, a sealed reaction system and exhaust gas recovery system should be utilized.

As for the experiment step 2.1.2, a more specific operation process is described as follows and added in the revised manuscript: First, a plastic beaker was filled with 200 ml deionized water, and 3.04 g of ZnF2•4H2O particles were taken and dispersed in the above deionized water by ultrasonic stirring until the solution became clear. Then, 5*5cm copper foil was washed with 50 ml of 1 mol/L dilute hydrochloric acid to remove surface impurities, and added to the above ZnF2•4H2O aqueous solution, which was mechanically stirred at room temperature for 3 hours. Finally, the modified copper foil was washed with deionized water and ethanol for 3-5 times to remove the alkaline salt on the modified copper foil, and the product obtained was named as ZnFx(OH)y@Cu. After the experiment, some lime water was prepared. The hydrolyzed HF was neutralized with lime water to produce non-toxic calcium fluoride precipitation.

  1. The SEI layer formation is mainly described by the 1st charge and discharge cycle is not disclosed in both main article and supplementary Materials?

Reply: Thanks for the reviewer’s important suggestion. In Figure 3a, the formation process of SEI has been described in detail.

The initial Coulombic efficiencies of the anodes based on bare Cu and ZnFx(OH)y@Cu are 97.1% and 98.5% respectively. The lower initial CE is related to the formation of a solid electrolyte interphase (SEI) layer during the first cycle. For the SEI layer on the anode of lithium metal battery, it is generally believed that the extremely reactive metal Li reacts with the anion in the electrolyte, and the reaction products (mostly insoluble) are deposited on the surface of lithium metal, forming a passivation film thick enough to prevent electrons from passing through. The formation of SEI layer will consume active Li in the battery, resulting in a Coulombic efficiency value lower than 100%. Such explanation has been added in the revised manuscript.

  1. The difference in coulombic efficiency for bare Cu(98.4%) and modified Cu(98.8%) is not much different which is basic claim of the study?(line 195)

Reply: CE mainly depends on the composition of electrolytes including salts and solvents. In this research, the working electrode and reference electrode were tested in the same electrolyte. Therefore, CE values are quite similar in two systems. For the working electrode, ZnF(OH) nanocrystals promote uniform Li nucleation and deposition. Therefore, the stability of CE values over cycling is better than that in the reference electrode.

  1. The authors explain the high performance based on 100 or 60 cycles and 519mAh/g on the other hand we have graphite (anode material) with thousand cycles with low cost?

Reply: Thanks for the reviewer’s comment. Lithium-ion battery based on transition metal oxide cathode and graphite anode can demonstrate long cycle life over thousands of cycles. However, the specific capacity and energy density can not meet the ever-growing demand of long-range electric vehicles. Sulfurized polyacrylonitrile cathode-based lithium sulfur batteries have much higher specific capacity therefore potentially much higher energy density. However, the intrinsic cycling stability is inferior to that of commercial lithium-ion battery, especially considering the cathode mass loading in this study is high.

Minor issues:

  1. Cyclic voltammetry is not mentioned in the research which is important

and corelated with the charge and discharge profile?

Reply: Thanks for the reviewer’s comment. CV is important for evaluation of electrochemical properties of battery materials. However, in this study, for both sulfurized polyacrylonitrile cathode and lithium metal anode, CV tests can not provide much more information beyond that galvanostatic charge/discharge tests have provided.

  1. The role of Se is not mentioned in the research?

Reply: Thanks for the reviewer's valuable comment. Se has better ionic/electronic conductivity than S, and can accelerate the conversion reaction kinetics, which has been demonstrated in previous research work (Nat. Commun. 10.1 (2019): 1-9). The above description has been added in the revised manuscript.

Round 2

Reviewer 3 Report

1. The ZnFx (OH) y compatibility is tested against the polyacrylonitrile based cathode material for the Lithium sulphur batteries. If this anode material is not compatible with other cathode material, then its use would be limited?

2. For the 2nd review answer, the HF handling/managing in the lab and as well as in industry is cumbersome and avoided. As you have a series of process to remove the toxicity of HF, made the simplicity of your method questionable?

3. The author did not pinout the 3rd review or the answer is unsatisfactory. Anyhow author explains the reason for the SEI layer formation but did not show the 1st charge/discharge cycle. On the other hand, Figure 3(a) does not explain the formation of SEI layer, it entails the information about CE only.

Note: Remaining answers found satisfactory.

Author Response

Reviewer’s Comments

  1. The ZnFx (OH) y compatibility is tested against the polyacrylonitrile based cathode material for the Lithium sulphur batteries. If this anode material is not compatible with other cathode material, then its use would be limited?

Reply: Thanks for the reviewer’s comment. ZnFx(OH)y coating layer is used to modify the surface property of copper foil and induce uniform Li nucleation, and its mass loading is tiny. In this research, lithium metal deposited on the modified copper foil is the anode, but not the ZnFx(OH)y coating layer. Lithium metal is generally compatible with the sulfurized polyacrylonitrile cathode and other cathode materials.

  1. For the 2nd review answer, the HF handling/managing in the lab and as well as in industry is cumbersome and avoided. As you have a series of process to remove the toxicity of HF, made the simplicity of your method questionable?

Reply: We thank the reviewer for this important comment. We agree with the reviewer that HF handling/managing in the lab and as well as in industry is cumbersome. The description on “simple synthesis” or “facile synthesis” has been removed in the revised manuscript.

  1. The author did not pinout the 3rd review or the answer is unsatisfactory. Anyhow author explains the reason for the SEI layer formation but did not show the 1st charge/discharge cycle. On the other hand, Figure 3(a) does not explain the formation of SEI layer, it entails the information about CE only.

Reply: Thanks for the reviewer’s comment. In this research, the Li/Cu cells were first cycled five times at 50 μA in the voltage range of 0-1 V (vs Li+/Li), followed by long-term cycling test at a current density of 0.5 mA cm-2 and a lithium deposition capacity of 1 mAh cm-2, which is described in the experimental section of the original manuscript. And it is a commonly used testing method in most of previous literatures. For the bare copper current collector, the above charge/discharge cycles at 50 μA is used to remove the surface oxides and other impurities which may bring about side reactions. And for the ZnFx(OH)y modified current collector, ZnFx(OH)y reacts with lithium over such charge/discharge cycles. Therefore, the initial cycle shown in Figure 3a demonstrates the reversibility of Li plating/stripping.

In terms of the SEI formation, the SEI composition is mainly determined by the electrolyte formulation. In this work, the introduction of ZnFx(OH)y may lead to the increase of LiF in SEI. Unfortunately, due to the COVID-19 lockdown here in our college, the XPS equipment for the analysis of SEI component is not available now. And we believe that the analysis of SEI component does not affect the main conclusion of this study.
